# Global Medicinal Use of Bats: A Systematic Literature and Social Media Review

Elaine S. Tackett, Tigga Kingston, Narges Sadeghmoghaddam and Abigail L. Rutrough *

Department of Biological Sciences, Texas Tech University, Lubbock, TX 79409, USA; elaine.tackett@ttu.edu (E.S.T.); tigga.kingston@ttu.edu (T.K.); narges.sadeghmoghaddam@ttu.edu (N.S.)
* Correspondence: abby.rutrough@ttu.edu

**Abstract:** The hunting of bats for food and medicine is one of the greatest threats to bat conservation. While hunting for consumption is the focus of increased attention, the specific medicinal uses of bats are poorly documented, limiting mitigation efforts. Here, we determine the distribution of bat hunting for food and medicinal use and characterize medicinal use practices. We systematically surveyed English-language scientific literature and social media platforms utilizing keywords and hashtags in 27 languages. We found 198 papers and 1063 social media posts from 83 countries and territories. Although use for food was more common, with 1284 unique reports from 71 countries, bats were used to treat 42 ailments of 11 human body systems across 37 countries (453 reports). Asthma was the most common ailment, distantly followed by kidney conditions. Ten organs or body parts of bats were used medicinally, with bat meat (flesh) and fluids (blood, bile, and oil) the most common. Understanding the effects and drivers of specific bat hunting practices will help guide conservation and public health efforts in the communities where bats are hunted. By pinpointing the ailments bats are being used for, outreach and alternative treatments can be introduced to replace the use of bats.

**Keywords:** bats; overexploitation; hunting; bushmeat; traditional medicine; conservation

## 1. Introduction

Overexploitation is one of the greatest threats to biodiversity [1], and biological resource use threatens nearly 38,000 species worldwide [2]. Bats are no exception, and many species are subject to intense hunting, particularly in the Old-World tropics [3,4]. Intentional biological use is listed as a threat to 125 of the 1332 bat species currently assessed by the International Union for the Conservation of Nature (IUCN) [2], and hunting is acknowledged as a direct driver of extinction of at least two species [5,6] and implicated in the loss of others (e.g., [7]). Bats are particularly vulnerable to any form of harvesting because they are long-lived and the majority of species give birth to just a single pup per year, resulting in low reproductive rates [8,9] that cannot compensate for current hunting levels [10–12], particularly in populations already declining as a result of habitat loss. With over 1450 species described [13], bats are an important component of global mammalian diversity and perform critical ecosystem services as agents of pest suppression, pollination, and seed dispersal [14–17]. Population declines consequently have ecological and economic consequences.

Broadly, bats are exploited for bushmeat and perceived medicinal value [4], and local motivations for exploitation may involve diverse cultural drivers (e.g., [18,19]). Hunting of bats for food is the most common focus of exploitation studies, yet the medicinal use of bats is geographically widespread, has a long history, and is likely a greatly underreported pressure on bat populations [4,20]. Riccucci [20] describes ancient medicinal use of bats, including a record from ~3400 BC of the use of bats in Egypt to treat skin disorders and vision impairment. Other ailments include mental health, pica, and hair growth, but

these treatments have not been quantified to determine their frequency of occurrence [20]. Mildenstein et al. [4] reported contemporary medicinal use of 18 species from five bat families, across 27 countries, primarily in Asia and Africa, but did not specify the ailments being treated. Despite this long documented historical and contemporary use, we could find no evidence of the effectiveness of bats for medicinal purposes. The sole possible exception was the development of a synthetic form of "Draculin"—the anticoagulant protein found in the saliva of the common vampire bat (*Desmodus rotundus*)—as a blood thinner to treat strokes. However, clinical trials were halted when the compound (desmoteplase) failed to demonstrate significant benefits, and treatment was not developed further [21].

We quantified the contemporary medicinal uses of bats and the geographical variation in food and medicinal practices. Data were collected from the IUCN Red List assessments, a systematic literature review, and social media keyword searches. We found reports of bats consumed for food from 71 countries and territories, and for medicine in 37. Across countries, we found that bats were being used to treat 42 medical conditions. These findings can be used to target healthcare and conservation initiatives in specific regions, increasing the likelihood of positive outcomes for both people and bats.

## 2. Materials and Methods

### 2.1. Scientific Literature Search

We searched the IUCN Red List on or before 14 October 2020 and compiled a list of all bat species that were listed under the threat classification 5.1.1. "Biological resource use–Hunting and trapping of terrestrial animals–Intentional use", 5.1.2. "Biological resource use–Hunting and trapping of terrestrial animals–Unintentional effects", and 5.1.4. "Biological resource use–Hunting and trapping of terrestrial animals–Motivation Unknown/Unrecorded". We reviewed all publications cited in the assessments and retained papers for further review if they included (1) information about bats, (2) a specific location (country), and (3) information about bat hunting, food, or medicinal use. Consumption of bats for food was distinguished from medicinal use in which bats (or parts of bats) were consumed or applied topically to treat or cure an ailment. Because it is hard to keep bats in captivity and they have low reproductive rates, they are not farmed [22]. Therefore, we assumed all bat products came from wild bats that had been hunted. We include poaching within hunting as we could not determine the legality of hunting nor the extent of law enforcement across countries. We collected location data and recorded whether bats were used for food or medicine, the specific ailments treated, the human organ system treated, the route of administration, and the part of the bat used. The data were categorized by IUCN region, with Asia additionally subdivided based on the subregions used by the World Commission on Protected Areas, and Afghanistan was included in West Asia [23,24]. This data acquisition resulted in 115 relevant papers published between 1974 and 2020.

To fill in gaps in the literature reported in the IUCN assessments, we conducted an all fields search in Web of Science (WoS). Key words were: "Bat Bushmeat" (*n* = 50), "Bat Cures" (*n* = 165), "Bat Exploitation" (*n* = 440), "Bat Hunting" (*n* = 695), "Bat Traditional Medicine" (*n* = 189), and "Bat Poaching" (*n* = 14). Search results included papers indexed up until 17 December 2021. We reviewed all search results and determined relevance using the same criteria as applied to the IUCN literature review, and excluded all papers where hunting occurred before 1970. This resulted in an additional 83 relevant papers published between 1975 and 2021 (Supplementary Materials Listing S1). The full literature search resulted in 198 relevant papers.

### 2.2. Social Media Search

To complement the literature review, we searched for evidence of bat hunting on social media. We searched 17 social media platforms in 23 languages from May 2020 to April 2021 (Supplementary Materials Table S1) and extracted 1034 posts dated from September 2007 to April 2021. The languages chosen were based on those that could be effectively translated by an available native speaker or using Google Translate. The searchers started

with larger global social media platforms, such as Instagram, Twitter, and Facebook, and then expanded to those more specialized to the language/region. Hashtags and keywords used in the searches started with broad concepts around bat hunting, such as '#batsoup' or 'bat hunting,' with additional search terms developed iteratively by examining terms that appeared in multiple posts (Supplementary Materials Table S1). Posts were retained for data extraction if they met the same criteria as the literature, specifically that they included (1) information about bats, (2) a specific location (country), and (3) information about bat hunting, food, or medicinal use.

We conducted a second social media search targeted at bat medicinal use in October 2021 to fill in gaps in the original dataset. We searched in 14 previously used and 4 new languages in the largest social media platforms—Instagram, Twitter, and Facebook. This search only included 18 languages rather than the original 23 because they were the most geographically distributed and could be translated via Google Translate, as we no longer had access to fluent speakers. The hashtags/keywords searched were broad to increase the potential of capturing any missed data and were specifically focused on finding evidence of medicinal use (Supplementary Materials Table S1). This search resulted in the extraction of an additional 29 posts from January 2020 to January 2021.

### 2.3. Data Extraction on Medical Use

The scientific literature and social media posts containing information on medicinal use were further investigated to determine: (1) the medical condition that was being treated, as described in the post or paper; (2) the human body system being targeted; (3) the part of the bat being used; and (4) the route of administration. We categorized the human body and its organ systems conventionally following Marieb and Hoehn [25]: nervous, integumentary, sensory, immune, respiratory, reproductive, urinary, cardiovascular, muscular, and digestive [25]. The part of the bat used is the organ, or other component of the bat, that was used in the treatment, such as the bat's liver, meat, guano, fluids, and hair (fur). Route of administration was defined as how the treatment was administered, such as ingestion or topical application. Posts or papers containing no information about the sections labeled "medicine" and "traditional medicine" were placed into unknown categories. A map of the medicinal use of bats was constructed using ArcMap 10.8 (ESRI, Redlands, California). In analyses, we defined a report as a unique location within a given source (paper or social media post).

## 3. Results

### 3.1. The Use of Bats

We found 1284 unique reports of bat consumption for food and 453 of medicinal use. There was wide variation in the spatial distribution of bat use for food and medicine. Bat consumption exclusively for food was found in 46 countries and territories, exclusive use for medicine in 12, and use for both in 25 (Supplementary Materials Table S2). Consumption for food was prevalent in regions of Asia, Africa, and Oceania, whereas medicinal use was most commonly reported in South Asia (where it exceeded consumption for food), Southeast Asia, and to a lesser extent, Africa and South America (Figure 1).

### 3.2. Medicinal Use of Bats

We quantified the specific ailments bats were believed to treat. Of the 453 reports of medicinal use, 190 reported a focal ailment, and we were able to attribute the ailment to a human organ system in 184 cases. We found bats were used to treat 42 medical conditions (Figure 2a) and 11 human organ systems (Figure 2b). The most common medicinal use was to treat asthma, followed by kidney conditions, cough, and paralysis. The publications and posts predominantly described the treatment of ailments affecting the respiratory system (83 observations), nervous system (27), and reproductive system (18) (Figure 2b).

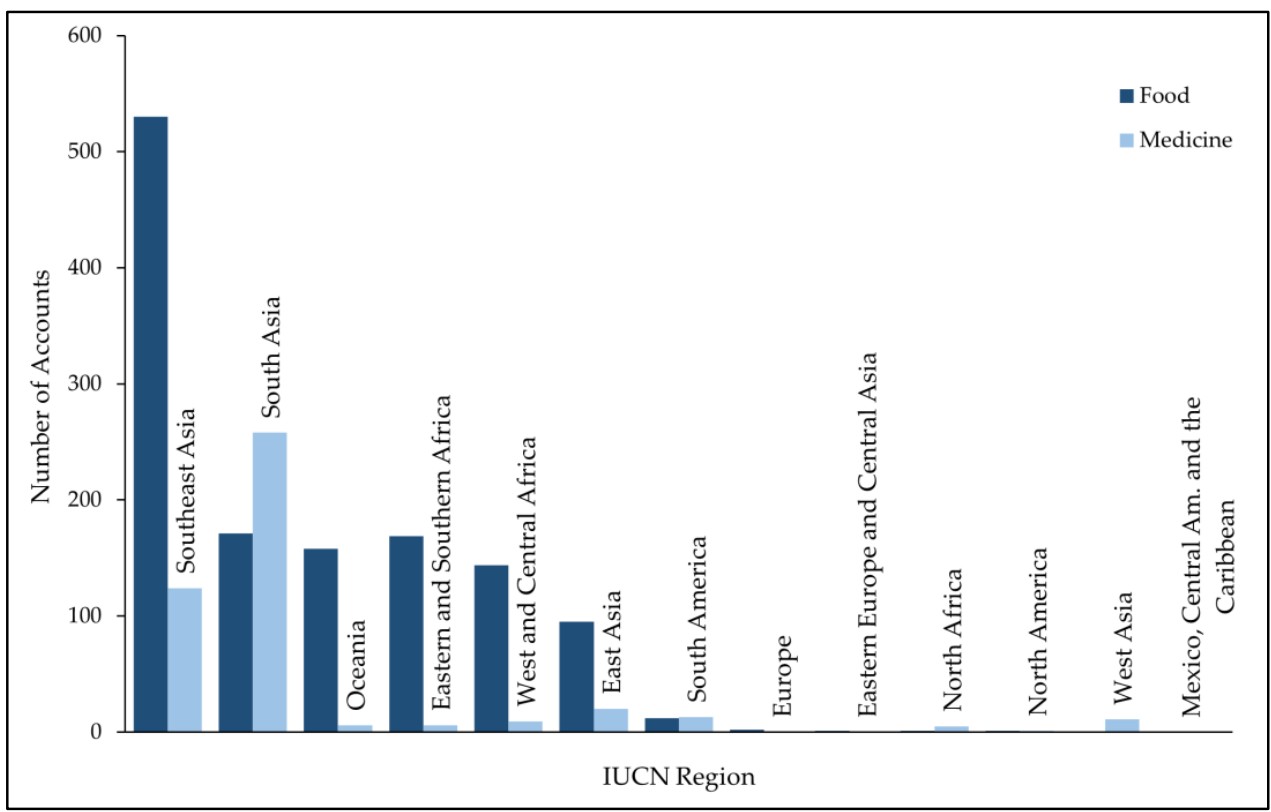

**Figure 1.** Comparison of bat usage for food (dark blue, *n* = 1284) and medicine (light blue, *n* = 453) measured as unique accounts of locations reported in the literature (1974–2021) and social media (September 2007–October 2021). Regions and subregions follow the IUCN classification system.

There were 108 records that reported the specific organs or body parts of bats used in medicine, and 104 records indicated how the bat treatment was administered (Figure 2c,d). We found 10 specific organs or bat body parts were used, and bats were only administered through ingestion and topical application. Bat meat was the most common body part administered followed by bodily fluids, such as blood, oil, and bile.

*3.3. Geographical and Taxonomic Distribution of Medicinal Use*

We found 36 bat species in six families, located in 14 countries were used for medicine (Figure 3, Table 1). Of the 36 species, 17 are frugivores, 17 are insectivores, 1 is nectarivorous, and 1 is a sanguinivore.

Bats are exploited for medicine in 8 out of 13 IUCN regions/subregions (Figure 4). Exploitation for asthma and other respiratory ailments was particularly common throughout Asia. Southeast Asia had the most reports of medicinal use and treated the greatest number of ailments (*n* = 23), followed by South Asia (*n* = 17), South America (*n* = 9), and West and Central Africa (*n* = 6).

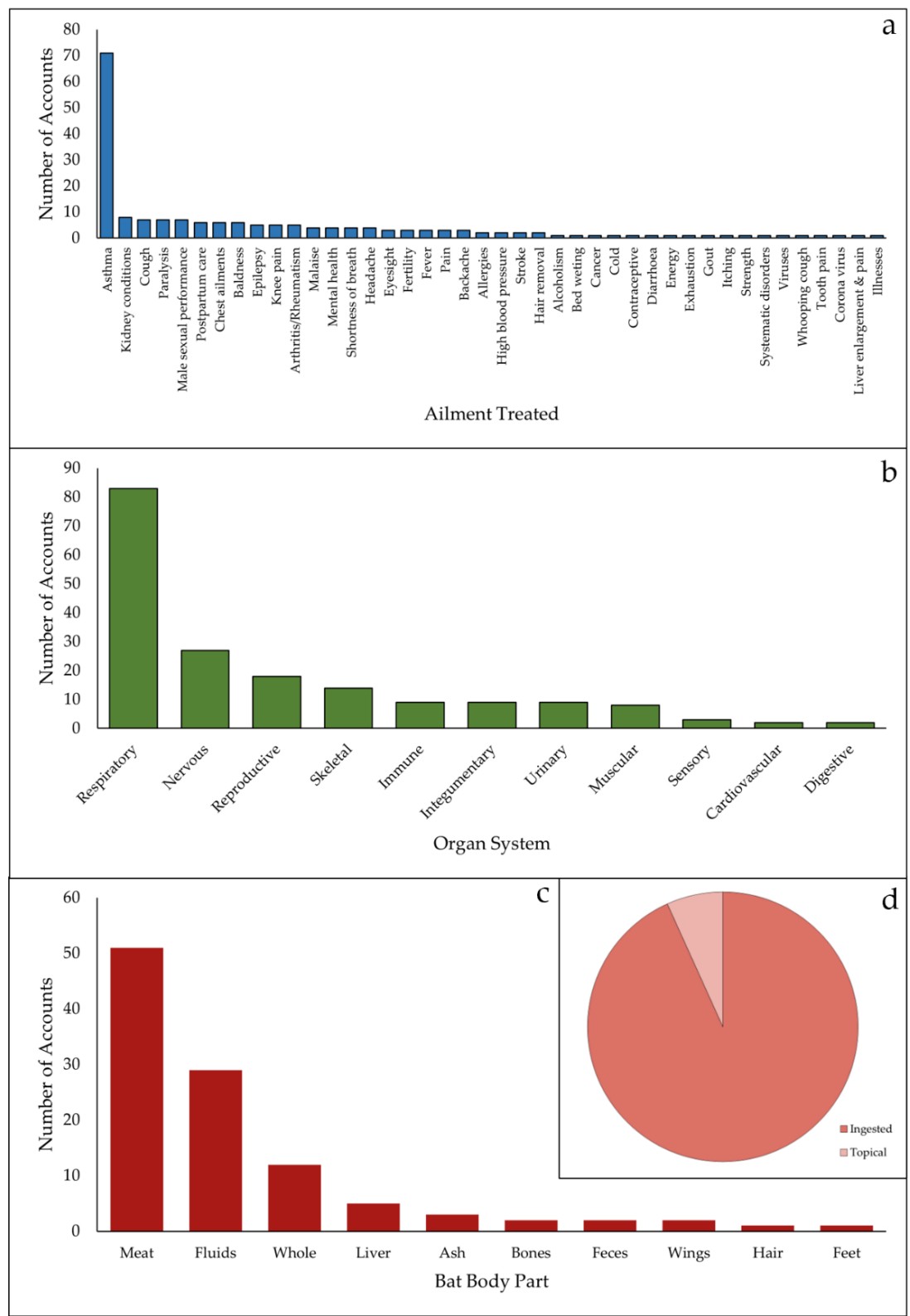

**Figure 2.** Specific ailments treated by bats (**a**) *n* = 190, human organ systems targeted (**b**) *n* = 184, the body part of bats used in medicine (**c**) *n* = 108, and route of administration of bats used in medicine (**d**) *n* = 104, measured as unique accounts reported in the literature (1974–2021) and social media (September 2007–October 2021).

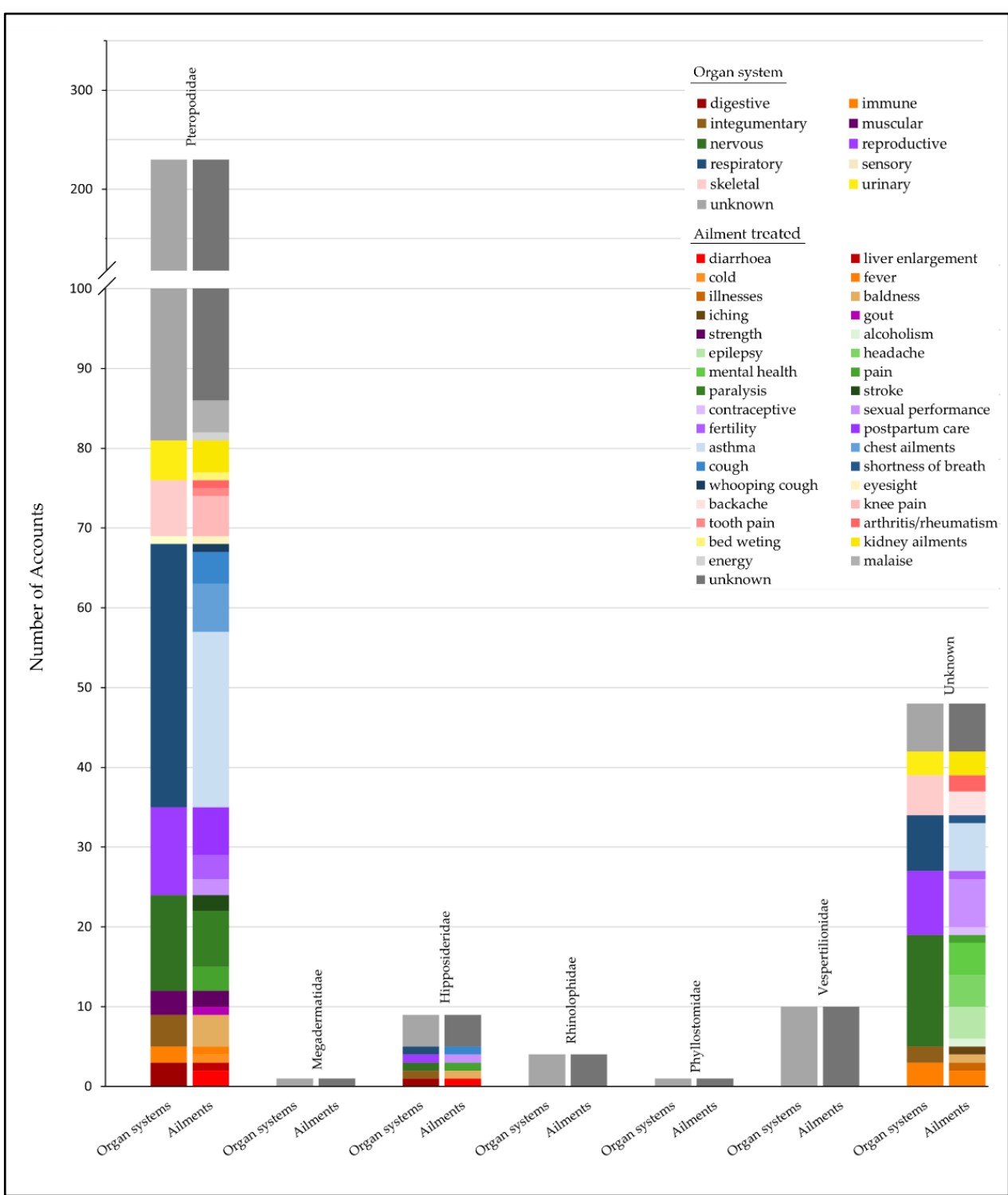

**Figure 3.** Medicinal use of bats by family. Human organ systems (left bar) aligned with the specific ailments that comprise them (right bar, *n* = 303). Data measured as unique accounts reported in the literature (1974–2021).

**Table 1.** Bat medicinal use by taxonomic group (*n* = 255), ailment treated, the human body system targeted, and the part of the bat used, as reported in the literature (1974–2021). The * indicates a species not found in Mildenstein et al., 2016 [4].

| Genus | Species | IUCN Status | Location | Ailment Treated | Organ System | Bat Part Utilized | Source |
|---|---|---|---|---|---|---|---|
| **Pteropodidae** | | | | | | | |
| *Unknown* | sp. | Unknown | Indonesia, Malaysia | Stroke, Paralysis, Baldness | Nervous, Integumentary | Fluids | Kasso and Balakrishnan 2013, Sheherazade and Tsang 2015 |
| *Acerodon* | *celebensis* * | VU | Indonesia | Unknown | Unknown | Unknown | Sheherazade and Tsang 2015 |
| *Cynopterus* | *sphinx* | LC | India | Eyesight, Asthma, Cough, Cold, Bed Wetting, Enlarged Liver, Tooth Pain | Respiratory, Digestive, Sensory, Immune, Urinary, Skeletal | Meat, Wings, Whole | Jugli 2020, Molur et al., 2002 |
| *Eidolon* | *dupreanum* * | VU | Madagascar | Whooping Cough, Baldness, Cough | Respiratory, Integumentary | Fluids | Fernández-Llamazares et al., 2018, Jenkins and Racey 2008 |
| *Eonycteris* | *spelaea* * | LC | India, Philippines | Diarrhea, Pain, Male Sexual Performance | Nervous, Digestive, Reproductive | Meat | Dovih 2015, Tanalgo et al., 2016 |
| *Epomophorus* | *crypturus* * | LC | Cameroon | Fertility | Reproductive | Whole | Van Cakenberghe and Seamark 2014 |
| | *labiatus* * | LC | Cameroon | Fertility | Reproductive | Unknown | Van Cakenberghe and Seamark 2014 |
| *Hypsignathus* | *monstrosus* * | LC | Cameroon | Fertility | Reproductive | Unknown | Van Cakenberghe and Seamark 2014 |
| *Latidens* | *salimalii* * | EN | India | Asthma | Respiratory | Unknown | Molur et al., 2002 |
| *Pteropus* | sp. | Unknown | Indonesia, Andaman and Nicobar Islands, Thailand | Asthma, Chest Ailment, Postpartum Care | Respiratory, Reproductive | Meat | Aul et al., 2014, Harrison et al., 2011, Suwannarong et al. 2020 |
| | *alecto* * | LC | Indonesia | Unknown | Unknown | Unknown | Sheherazade and Tsang 2015 |
| | *faunulus* | EN | Andaman and Nicobar Islands | Strength, Asthma | Muscular, Respiratory | Meat, Bones | Aul et al. 2014, Aul and Vijaykumar 2003 |
| | *hypomelanus* * | NT | Malaysia, Thailand | Asthma | Respiratory | Unknown | Aziz et al., 2012, Mickleburgh et al., 1992 |
| | *lylei* * | VU | Thailand | Energy | Unknown | Fluid | Weber et al., 2015 |

**Table 1.** *Cont.*

| Genus | Species | IUCN Status | Location | Ailment Treated | Organ System | Bat Part Utilized | Source |
|---|---|---|---|---|---|---|---|
| **Pteropodidae** | | | | | | | |
| | *medius* | LC | India, Nepal, Bangladesh, Pakistan | Fever, Pain, Asthma, Paralysis, Knee Pain | Immune, Nervous, Respiratory, Skeletal | Hair, Fluids | Mahmood et al., 2010, Mickleburgh et al., 1992, Molur et al., 2002, Molur et al., 2007, Neupane et al., 2016, Openshaw et al., 2016, Rahmatullah et al., 2013 |
| | *melanotus* * | VU | Andaman and Nicobar Islands | Strength, Asthma | Muscular, Respiratory | Meat, Bones | Aul and Vijaykumar 2003 |
| | *vampyrus* * | NT | Indonesia, Malaysia | Rheumatism/Arthritis, Cough, Gout, Asthma, Malaise, Kidney Ailments, Chest Ailments, Postpartum Care | Respiratory, Muscular, Skeletal, Reproductive, Urinary | Meat, Fluids | Fujita and Tuttle 1991, Harrison et al., 2011, Masy'ud et al., 2020, Rahman 2010 |
| *Rousettus* | *amplexicaudatus* * | LC | Philippines | Unknown | Unknown | Meat | Tanalgo et al., 2016 |
| | *leschenaultii* | NT | India | Pain, Diarrhea, Male Sexual Performance | Nervous, Digestive, Reproductive | Unknown | Dovih 2015, Molur et al., 2002 |
| | *madagascariensis* * | VU | Madagascar | Baldness, Cough | Integumentary, Respiratory | Fluids | Fernández-Llamazares et al., 2018 |
| **Megadermatidae** | | | | | | | |
| *Megaderma* | *spasma* | LC | India | Unknown | Unknown | Unknown | Molur et al., 2002 |
| **Hipposideridae** | | | | | | | |
| *Hipposideros* | *armiger* * | LC | India | Diarrhea, Pain, Male Sexual Performance | Nervous, Digestive, Reproductive | Unknown | Dovih 2015 |
| | *diadema* * | LC | Philippines | Unknown | Unknown | Meat | Tanalgo et al., 2016 |
| | *galeritus* * | LC | Sri Lanka | Unknown | Unknown | Unknown | Molur et al., 2002 |
| | *speoris* | LC | India | Unknown | Unknown | Unknown | Molur et al., 2002 |
| *Macronycteris* | *commersoni* * | NT | Madagascar | Baldness, Cough | Integumentary, Respiratory | Fluids | Fernández-Llamazares et al., 2018 |
| **Rhinolophidae** | | | | | | | |
| *Rhinolophus* | *fumigatus* * | LC | Senegal | Unknown | Unknown | Unknown | Van Cakenberghe and Seamark 2014 |

**Table 1.** *Cont.*

| Genus | Species | IUCN Status | Location | Ailment Treated | Organ System | Bat Part Utilized | Source |
|---|---|---|---|---|---|---|---|
| **Pteropodidae** | | | | | | | |
| | *luctus* * | LC | India | Unknown | Unknown | Unknown | Molur et al., 2002 |
| **Phyllostomidae** | | | | | | | |
| *Desmodus* | *rotundus* * | LC | Brazil | Unknown | Unknown | Unknown | Rocha et al., 2021b |
| **Vespertilionidae** | | | | | | | |
| *Eptesicus* | *serotinus* * | LC | India | Unknown | Unknown | Unknown | Molur et al., 2002 |
| *Hesperoptenus* | *tickelli* * | LC | India | Unknown | Unknown | Unknown | Molur et al., 2002 |
| *Myotis* | *capaccinii* | VU | North Africa | Unknown | Unknown | Unknown | Paunović 2016 |
| | *emarginatus* | LC | North Africa | Unknown | Unknown | Unknown | Piraccini 2016, Van Cakenberghe and Seamark 2014 |
| | *punicus* | DD | North Africa | Unknown | Unknown | Unknown | Van Cakenberghe and Seamark 2014 |
| | *mystacinus* | LC | West Africa | Unknown | Unknown | Unknown | Van Cakenberghe and Seamark 014 |
| *Nyctalus* | *montanus* | LC | Nepal | Unknown | Unknown | Unknown | Molur et al., 2002 |
| *Pipistrellus* | *ceylonicus* * | LC | South Asia | Unknown | Unknown | Unknown | Molur et al., 2002 |
| | *coromandra* * | LC | India | Unknown | Unknown | Unknown | Molur et al., 2002 |

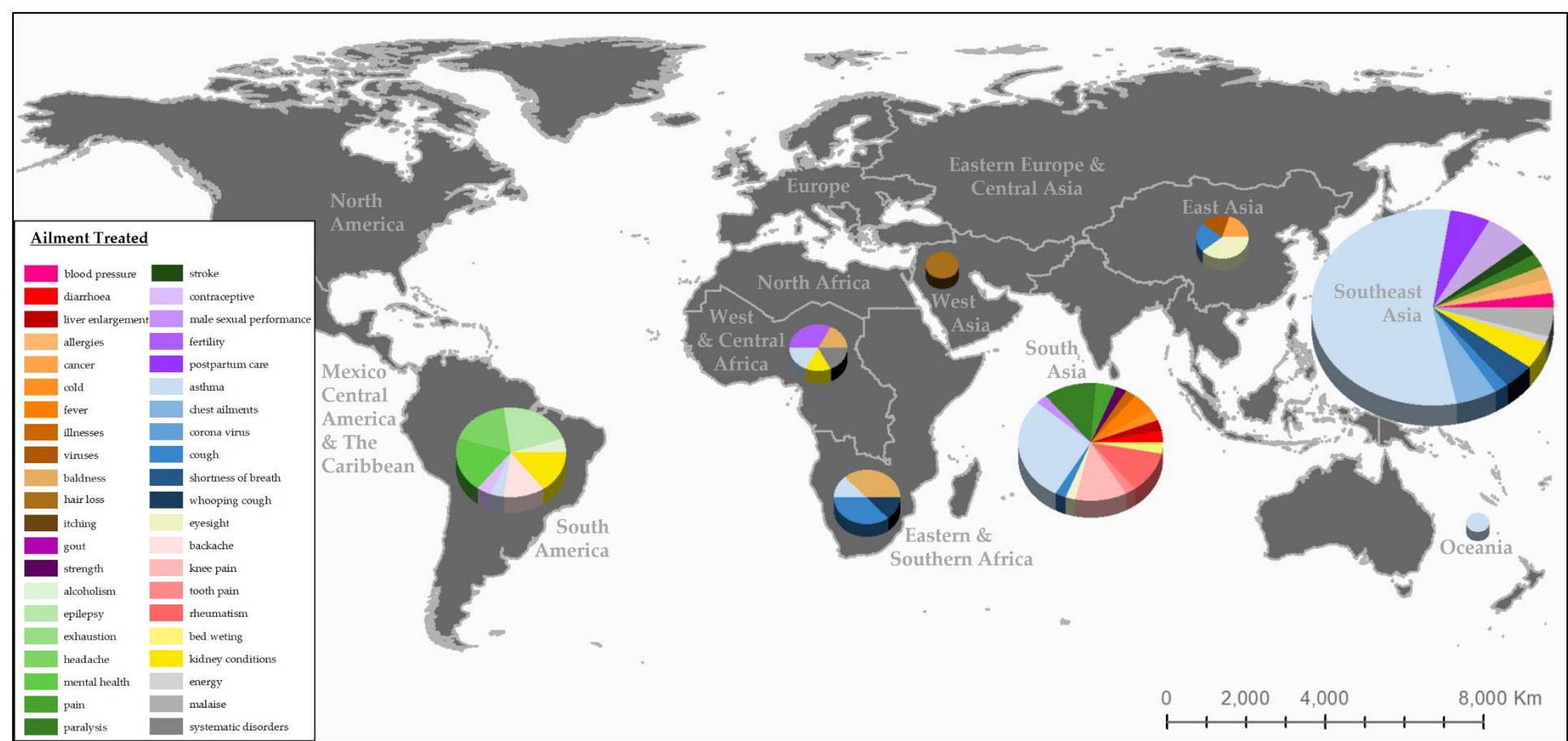

**Figure 4.** Global variation in type and quantity of ailments treated with bats (*n* = 188), measured as unique accounts reported in the literature (1974–2021). Size of pie charts reflect the number of data points in that region. Regions and subregions follow the IUCN classification system.

## 4. Discussion

We found evidence that people from 83 countries and territories use bats for food and/or medicine (Supplementary Materials Table S2). Similarly, we found a great diversity in the types of ailments being treated with bats. We found that bats are being used to treat 42 different ailments in 37 countries; the ailments covered a wide range of physical conditions, such as asthma, pain, and mental health disorders. We expected that our findings would align with those of Mildenstein et al. [4], who reported medicinal use in 18 species across 5 families, comprising 6 frugivores and 12 insectivores bats, located in 27 countries. Although there is some overlap, we did not gather any data about the family Emballonuridae, though widely distributed, and did not collect any records of *Eidolon helvum, Hipposideros lankadiva, Hipposideros pomona, Megaderma lyra, Myotis nattereri, Taphozous theobaldi,* or *Pteropus vetulus.* However, we did added 25 species (designated with an * in Table 1), bringing the total number of species for which there are reports of medicinal use to 43 (observations only reported to genus level are excluded from this count).

Typically, previous work on bat hunting only distinguished between exploitation for bushmeat or medicine. In particular, reports of medicinal use in scientific literature lack detail and are frequently referred to only as "traditional medicine" or "medicine" (e.g., [26–28]). Of the 453 reports of medicinal use we retrieved, only 190 reported a focal ailment. One critical finding of this work is the need to better document the specifics of bat hunting for medicinal use. Documenting the specific ailments that bats are used to treat can help guide conservation and public health interventions, better utilizing limited resources. Efforts can be made to promote or supply accepted conventional medicines to treat ailments, although this may require sensitivity to cultural norms. Capitalizing on changing perceptions may also be effective. For example, asthma sufferers in Indonesia commonly use bats to as a treatment, but we found several posts in which Indonesian social media commentators highlighted that this is ineffective and instead promoted the use of inhalers. Thus, we recommend future authors include specific details of the human ailment being treated, part of the bat used, and the route of administration, rather than labeling them as merely medicine or traditional medicine.

For many of the ailments, bats were used as a treatment because there was either minimal or no access to other medical treatments [20]. However, the beliefs behind many of these treatments are not well documented, and there is a call to document not only how bats are used but why [29]. For example, one interesting explanation for the belief that bats treat mental health conditions comes from the idea that bats' ability to fly at night indicates their ability to orient themselves and thus they can cure patients who experience mental disorientation [20]. We agree with Jaroli [29] that further research should be conducted to determine the origins of the beliefs behind the medicinal use of bats.

The treatment of asthma with bats has been well documented in the literature (e.g., [4,19,20]), however an unexpected outcome was the number of other conditions that were treated with bats (Figure 2a). At the global scale, some conditions even appear contradictory, such as bats being used to treat baldness in parts of Sub-Saharan Africa and Southeast Asia, but for hair removal in West Asia, or as a contraceptive in South America and to promote fertility in West and Central Africa. We reiterate that we found no evidence that bats have any medicinal effect.

There were many limitations to the social media research we conducted. Although we were able to access fluent speakers of 10 languages, and could reliably translate several more with Google Translate, many languages are not represented in our study. Mainland Southeast Asian languages and Russian are entirely missing from our social media search, whereas Chinese was only translated by a non-fluent speaker. From our literature review, it would be fruitful to search social media of Thailand, Vietnam, Russia, and China, which were not available to us, for the specifics of medicinal use. In addition, although the use of social media is globally widespread, there are gaps in access to mobile phone technology and data coverage, particularly in remote rural areas [30], where medicinal use of bats might be anticipated. Limitations of the scientific literature include unequal spatial distribution

of research due to unsafe working environments, under-sampling of certain regions due to political instability, and a lack of access to certain people/regions. Some regions may also have less data due to negative beliefs. For example, in Pakistan bats are believed to be impure, evil, or a form of devil, so people may be less willing to reveal if or how they use bats [31].

Hunting of bats for food has gained increased attention in light of the COVID-19 pandemic and Ebola epidemics. However, their exploitation for traditional medicine remains underexplored. Here we quantified the medicinal use of bats using available data sources (Supplementary Materials Table S3). Future work should aim to fill in gaps in the published scientific literature, especially in areas where limited internet access restricts social media use. An increased understanding of this human/wildlife interface will better inform research, public health and bat conservation.

**Supplementary Materials:** The following are available online at https://www.mdpi.com/article/10.3390/d14030179/s1. Listing S1: Results of the Quantitative Scientific Literature Search (*n* = 198). Table S1: Languages (*n* = 27), key terms (*n* = 720), and social media sites (*n* = 20) searched for evidence of bat exploitation for food and medicine. Search conducted June 2020–October 2021. Table S2: Bat use by country (*n* = 83), measured as unique accounts of locations reported in the literature (1974–2021) and on social media (September 2007–October 2021). Table S3: Delocalized data on the medicinal use of bats extracted from scientific literature and social media (*n* = 398).

**Author Contributions:** Conceptualization, E.S.T. and N.S.; methodology, E.S.T. and A.L.R.; software, A.L.R.; validation, E.S.T. and A.L.R.; formal analysis, E.S.T. and A.L.R.; investigation, E.S.T. and N.S.; resources, A.L.R. and T.K.; data curation, E.S.T. and N.S; writing—original draft preparation, E.S.T. and N.S.; writing—review and editing, E.S.T., N.S., A.L.R. and T.K.; visualization, E.S.T., and A.L.R.; supervision, A.L.R. and T.K.; project administration, A.L.R. and T.K. All authors have read and agreed to the published version of the manuscript.

**Funding:** This work was supported by the National Science Foundation Graduate Research Fellowship Program (DGE 2140745) to ALR, and National Science Foundation AccelNet Award Number 2020595 to TK at Texas Tech University. Any opinions, findings, conclusions. or recommendations expressed in this work are those of the author(s) and do not necessarily reflect the views of the National Science Foundation.

**Institutional Review Board Statement:** Not applicable.

**Data Availability Statement:** Data to country level are available in Supplementary Materials Table S2 and delocalized data are available in other supplementary materials. Due to the sensitive nature of these data, the full dataset is only available by request.

**Acknowledgments:** We thank L. Alvarez and G. Santana Lamboy for their contributions to data collection and early idea development. We are also thankful for the comments and revisions of A. Batrice, K. Paul, M. Sharma, L. Alvarez, K. Bent, and M. Ochoa for their comments and B. Tanley for aid in creating initial figures. We thank D. Gomes and two anonymous reviewers for their constructive comments.

**Conflicts of Interest:** The authors declare no conflict of interest.

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
