# Peer review of "Global Medicinal Use of Bats: A Systematic Literature and Social Media Review"

_diversity, doi:10.3390/d14030179_

Round 1
Reviewer 1 Report
This manuscript is a descriptive exploration of how bats are used around the world. The authors distinguish between food uses and medicinal uses. Overall, the manuscript is easy to follow and is to the point. There are some formatting issues and spelling errors throughout that the authors should remedy. The figures aren't the most visually-appealing, but they convey the necessary information.
The authors provide some of the data that they've collected, but mention that some of it is sensitive. I'd ask that the authors revisit this and assess if the data really are sensitive. Data on endangered species are not necessarily sensitive.
I also wonder about the distinction (or lack thereof) between hunting and poaching. Hunting in North America is generally thought to be legal, and managed in a sustainable way, whereas poaching is the legal, non-managed version whereby people take what they want. The authors mention using hunting and poaching as keywords, but mostly refer to harvest or take as "hunting" in the body of the text. As the text currently reads, it can be construed as hunting is a bad (or negative) thing, whereas hunting is currently how the United States funds most of its conservation work. Is there any way to weave this into the narrative (e.g. that hunting is not controlled / managed in many of these places such that sustainable take of these wildlife is not possible as it is elsewhere)?
I also offer some minor suggestions below:
Line 38: "offtake"? Not sure what this means here.
Line 52: "We hypothesized that there will be variation in the use of bats globally..." This is a prediction, not a hypothesis. Nonetheless, I do not think this sentence adds anything useful. The segue between the sentences before and after this one is clear and thus this one is not needed - consider removing it. It feels like an attempt to force a hypothesis into the introduction.
Lines 57-60: these are results, presented in the introduction. I haven't ever seen a paper written this way. This isn't necessarily problematic, but just thought I would point out that it is a little odd.
Line 75: period should come right after reference "[21]" (there is an additional space), check throughout by searching " ." throughout the document.
Line 83: extra space after "assessments,". Check throughout document.
Line 89: "occored" is a misspelling.
Line 90: "lerature" is a misspelling.
Line 103: "Relevance was determined and data were collected using the same criteria as in the literature search." - It is unclear exactly what this means. Can you explain how relevance was determined? Is it if they "included (1) information about bats, (2) a specific location (country) and (3) information about bat hunting, food or medicinal use."? Either way, perhaps you can make this link more apparent.
Line 130: "soource" is a misspelling.
Figure 1: y axis is "Number of Locations", but they are all within a region. Is this "Number of countries" or "Number of publications" that mention consumption? Towns/villages? Locations is vague here in this figure (i.e. not described in the caption). I would consider making this more obvious (many readers look at figures but don't bother with text, it may be helpful to remember this).
Figure 1: It would also make this figure more easily interpretable if there was a legend for the black and grey bars (although I recognize that it is mentioned in the caption).
Line 157: The wording here is odd around points. It seems weird to say that the __ system "had" X data points. The system didn't have any points, rather you collected X data points relating to said system, right? Why not, "there were 34 publications (or studies) that mentioned the ___ system" or "34 observations"?
Figure 2: Again, "Number of Locations" is confusing / vague. You might consider adding "Human" to "Organ System" x axis title for Figure 2b.
Why does the figure numbering in the manuscript go from Figure 2 to Figure 4?
Line 176-177: Again, this is a prediction (that there is variation in use of bats globally), not a hypothesis. I also don't think it is a very useful or interesting prediction. We expect *some* variation in just about everything globally, don't we? I would remove all mentions of hypothesis here. This is an exploratory or descriptive paper, not one where you knock down alternative hypotheses to explain some phenomenon.
Line 187: "This was clear in our research with 300 plus data points removed most analyses" - I am not sure what you mean by this statement. It is confusing and doesn't appear to be worded correctly. Can you clarify what you mean?
Line 190: "uilizing" -> "utilizing"
Lines 189-193: "Understanding the specific medical uses of bats can help guide conservation and public health interventions, better uilizing limited resources. Thus, we recommend future authors include specific detail on on the human ailment being treated, part of the bat used, and mode of administration, rather than labeling them as merely medicine or traditional medicine."
- Can you explain how understanding these things will aid in conservation? I don't think you've convinced me as a reader how knowing this will stop the overexploitation of these species. Your next sentence goes in a direction that could be helpful "bats were used as a treatment because there was no, or reduced, access to other medical treatments" -- does knowing what people are using bats for give us the potential to bring those regions other (e.g. western) medicines that can replace the use of bats?
Line 207: "fluint" -> "fluent" (again on line 210).
Line 221: "ingreased" -> "increased"
Line 224: "interent" -> "internet"
Line 227: "conservations" -> "wildlife conservation"
Line 241: "Due to the sensitive nature of these data, the full dataset is only available by request." Are these data revealing of endangered species locations? Or do they contain personally identifiable information from social media sites? What is sensitive about the data?
Line 243: sentence ends unexpectedly (with "to").
I really appreciate the large appendices, but I wonder if you can provide more data than appendix C provides, while still keeping any sensitive data confidential?
Line 1102: The manuscript mentions "References", but then has a table that is maybe the same as Appendix C repeated?
References are not consistent. Some author names are all capitol letters, some tabs introduced, some links are underlined, others not.
- Dylan Gomes
Author Response
Please see attached file, thank you for your feedback.

Reviewer 2 Report
Given the increasingly negative opinion of bats globally in the shadow of the current pandemic, this paper makes an important contribution to ongoing efforts to protect bat populations that are declining globally. The survey of medicinal uses is novel and some of the results are surprising (e.g., the use of bats for mental health). If possible, it would be interesting to see medicinal uses by taxonomic group (I suspect most of the medicinal use was utilizing larger pteropodids?); however, I understand that this level of detail may not have been available.
The paper is well written with only a few typos that a spell check should fix.
Author Response

(The authors gave the same response as above.)

Reviewer 3 Report
Dear Authors,
I enjoyed reviewing this very interesting manuscript. I think it fills a gap in the understanding of threats to endangered species, in this case to bats.
However, I am missing a clear statement about "evidence of medical effect of using bats as medicine". Is there any? At least you could say that there is none reported in the scientific literature or that you could not find it if you don't want to doubt directly. I think it's very important that you also transport the message that bats are no valid medicine and have no proven medical effect like a pharmaceutical drug. Thus, no reason to kill bats for "medical purpose". You can either state it already in the introduction or at least bring it into the discussion (or both).
In relation to this it would have been also interesting to look in the literature / social media post whether the treatment was successful (and if this is officially confirmed by a "real" doctor or just a belief).
minor comments:
- I think there is a problem with the numbering of the references since it starts with "0" in the reference list but in the manuscript text it starts with 1 (and the references also fit if it would start with 0 in the manuscript text).
- line 36/37: "...long-lived and typically give birth to just a single pup a year..." There are bat species that regulary give 2 pups. You can write e.g. "mostly one pub" or "typically one or two pups"
- line 37/38: "...resulting in low reproductive rates that do not support current offtake levels..." - "support" sounds a bit weird in this sense. Please rephrase.
- about the graphs: your graphs are simple and generally ok but there are much fancier plots to visualize your results. I recommend to use such modern vizualization (e.g. a map with quantitative bars / pies / circles). It would improve illustration/vididness of your very interesting findings.
- line 185/186: I think using "evident" here could cause confusion. Please replace or rephrase.
- line 189/190: "...Understanding the specific medical uses of bats can help guide conservation and public health interventions..." and also show the significance of the problems with misbeliefs of the medical value of endangered species
- line 190: typo? "uilizing"
- line 191/192: "...we recommend future authors include specific detail on on the human ailment being treated, part of the bat used, and mode of administration,..." Would you also recommend to include details whether there was an evident effect or not? If yes, please add to the sentence.
- line 199 ff: you only comment on the "mental health ailment". But are the treatments for other diseases more valid? I don't think so. You should explain it more general why bats are used as medicine (misbeliefs, curious derivations) and then you can give two or three most prominent examples.
- line 200: Point missing in the end of the sentence.
- line 221 ff: In these conclusive remarks you could also add a statement where bats have indeed a high medical value - in scientific medical research.
Author Response

(The authors gave the same response as above.)
